# Hospitalization and ambulatory costs related to breast cancer due to physical inactivity in the Brazilian state capitals

**Diego Augusto Santos Silva** [ID]*

Research Center in Kinanthropometry and Human Performance, Sports Center, Federal University of Santa Catarina, Florianopolis, Brazil

* diegoaugustoss@yahoo.com.br

## Abstract

The aim of this study was to estimate the hospitalization and ambulatory costs related to breast cancer due to physical inactivity in the female population from Brazilian capitals over a three-year period (2015 to 2017). This study was carried out with data from the Brazilian health system and had as metrics incidence of breast cancer, total and standardized rate hospitalizations by breast cancer, hospitalization and ambulatory costs by breast cancer and prevalence of physical inactivity. The Population Attributable Fraction (PAF) calculation was used. The total hospitalization cost by breast cancer in women aged $\geq$ 20 years in Brazil from 2015 to 2017 was US$ 33,484,920.54. Of this total, US$ 182,736.76 was due to physical inactivity. Outpatient expenses related to breast cancer in the Brazilian female population from 2015 to 2017 was US$ 207,993,744.39. Of this total, US$ 1,178,841.86 was due to physical inactivity. Outpatient and hospitalization expenses were higher in the states of Southeastern, Southern and Northeastern regions. Physical inactivity has contributed to the high number of hospitalizations for breast cancer in Brazil, which resulted in economic burden for health services (inpatient and outpatient) of more than US$ 1,300,000.00 from 2015 to 2017.

**Data Availability Statement:** All data are publicly available by Brazilian Unified Health System accessed via Department of Informatics

## Introduction

Breast cancer is the type of cancer most commonly diagnosed in the female population and in 2015, it accounted for 523,000 deaths worldwide [1, 2]. Breast cancer has multifactorial cause in which genetic and lifestyle aspects stand out [1]. Among modifiable lifestyle aspects, physical inactivity plays an important role in the prevention and treatment of this neoplasm because the regular practice of physical exercise is associated with lower concentrations of female sex hormones and lower levels of body fat [1]. Both factors (high concentrations of female sex hormones and high levels of body fat) are associated with increased risk of breast cancer, especially in postmenopausal women [1, 2]. Epidemiological surveys on breast cancer mortality due to physical inactivity estimated 29,605 deaths in 1990 and 46,720 deaths in 2015 around the world [3].

(DATASUS) of free access and charge: https://datasus.saude.gov.br/informacoes-de-saude-tabnet/. The authors did not have any special access privileges that others would not have.

**Funding:** The author(s) received no specific funding for this work.

**Competing interests:** The authors have declared that no competing interests exist.

In addition to mortality estimates presented in literature, the analysis of the economic burden of breast cancer can serve to assess public policies and for governments to prioritize preventive measures in order to reduce the economic cost of this disease [4]. Data from Spain published in 2020 reported that the total cost of breast cancer in that country over a five-year period was € 469,92.73 (US$ 557,43.01) [4] (€ 1.00 = US$ 1.19 in June 18, 2021). In Mexico, cost projection of US$ 245 million was estimated for procedures related to breast cancer throughout the lives of women born in 2012 [5]. In the United States of America, 2007 data have shown total cost of US$ 12.2 billion with procedures related to breast cancer [6]. In Brazil, cost of approximately US$ 5.8 million was estimated in 2013 for hospitalizations due to breast cancer [7].

In addition to information on the global economic burden of the disease, which can be analyzed by metrics related to cost related to hospitalizations, medicines and procedures in general [6, 7], studies have prioritized making estimates by risk factors [8]. These risk factor estimates allow managers and society to become aware of how much each action in a specific risk factor would result in savings. A systematic review developed with studies published until 2014 found 24 articles that estimated the cost of physical inactivity related to various non-communicable diseases and reported that physical inactivity had high economic impact for the health sector, regardless of economic analysis metric [8]. Ding et al. [9] estimated costs of physical inactivity related to various non-communicable diseases around the world and found that in 2013, this cost was US$ 53.8 billion, of which US$ 31.2 billion were paid by the public sector. In relation to breast cancer, a survey highlighted the impact of physical inactivity on hospitalizations for breast cancer and estimated an approximate expenditure of US$ 1.2 million in a single country [7].

Brazil has continental dimensions, with evident social and economic discrepancies among states [3, 10]. These discrepancies are reflected in living conditions, lifestyles and in the quality and access to health services [10], which, in turn, reflect in the amount of care for breast cancer and the cost of each state with the treatment of the disease [3]. Thus, the estimate of the economic burden of breast cancer due to physical inactivity by Brazilian state can provide information on how each state has faced the problem of breast cancer related to physical inactivity and also can provide information on the health inequality in Brazil.

The aim of this study was to estimate the hospitalization and ambulatory cost related to breast cancer due to physical inactivity in the female population from Brazilian capitals over a three-year period (2015 to 2017).

## Materials and methods

### Study design

This is a cost-of-illness study to estimate the direct costs of breast cancers attributable to lack of physical activity from the perspective of the Brazilian health services. The study's analysis units are the states of Brazil. This study was carried out with data from the Brazilian Unified Health System accessed via Department of Informatics (DATASUS) of free access and charge. For all Brazilian states, the quality of data from DATASUS is considered high and close to high-income countries [11, 12]. In addition, this study used information collected by the Ministry of Health of Brazil entitled "Surveillance System of Risk and Protection Factors of Non-communicable Disease by Telephone Survey—VIGITEL". This system aims to monitor health indicators of the Brazilian population aged ≥ 18 years through telephone survey and was carried out in 26 Brazilian capitals and the Federal District. Free and informed consent was obtained from all participants at the time of data collection. The VIGITEL study in Brazil protocol was approved by National Ethics Committee on Research with Human Beings (CONEP/

BRAZIL), and has been conducted in full accordance with ethical principles, including provisions of the World Medical Association Declaration of Helsinki (Ethical Application Ref: 65610017.1.0000.0008).

## Hospitalizations and ambulatory costs data

For the present study, malignant breast cancer was characterized according to the latest International Classification of Diseases, 10th Revision (ICD-10). ICD-10 categories for malignant breast cancer analyzed were C50, C50.0, C50.1, C50.2, C50.3, C50.4, C50.5, C50.6, C50.8, C50.9 [13].

From the access to the DATASUS system (i.e, SIH—Hospital Admission System), information from years 2015, 2016 and 2017 on the total number of hospitalizations and the cost of these hospitalizations in the Brazilian female population aged ≥20 years was extracted [14, 15]. Outpatient procedures related to breast cancer were extracted from Outpatient Information System from DATASUS (SIA/SUS) of the years 2015, 2016 and 2017. The total cost of outpatient procedures related to breast cancer in 2015, 2016 and 2017 were related to Brazilian capitals and the Federal District. For outpatient estimates, the procedures recommended by DATASUS were considered. All outpatient procedures related to breast cancer with the respective DATASUS codes can be found in the S1 Table.

From values of years 2015, 2016 and 2017, the mean cost of hospitalizations in these three years (with standard deviation estimate) and the total cost were calculated from the sum of annual values. For international comparison, estimates in reais (R$—Brazilian currency) were converted into US dollars (US$). For this purpose, the mean value of the dollar quotation in the three years of the survey was considered (US$ 1.0 = R$ 3,33) [16].

## Prevalence of physical inactivity

For estimates on the prevalence of physical inactivity in Brazilian women (aged ≥20 years), national surveys of the Surveillance System for Risk and Protective Factors for Chronic Diseases by Telephone Survey (VIGITEL) for years 2015, 2016 and 2017 were used [17–19]. Such a system was implemented in Brazil in 2006 and is consolidated as a health surveillance and management system. The sampling procedures used in this survey aim to obtain, in each capital of the 26 Brazilian states and the Federal District, probabilistic samples of the population of adults living in households served by at least one fixed telephone line. The system establishes minimum sample size of approximately 2 thousand individuals in each city to estimate, with 95% confidence coefficient and maximum error of two percentage points, the frequency of any risk factor in the adult population. Maximum errors of three percentage points are expected for specific estimates, according to sex, assuming similar proportions of men and women in the sample [17–19]. Smaller samples are accepted in locations where fixed telephone coverage is less than 40% of households and where the absolute number of households with a telephone line is less than 50 thousand. In this case, estimates for the adult population will have maximum error of three percentage points, being of four percentage points the same error for sex-specific estimates [17–19].

In 2015, 54,174 individuals were evaluated, of which 33,806 were women (n = 730 women aged 18–19 years; n = 33,076 women aged ≥ 20 years). In 2016, 53,210 individuals were evaluated, 32,952 of which were women (n = 757 women aged 18–19 years; n = 32,195 women aged ≥ 20 years). In 2017, 53,034 individuals were evaluated, of which 33,530 were women (n = 674 women aged 18–19 years; n = 32,856 women aged ≥ 20 years).

In the present study, estimates of physical inactivity were considered in all domains (leisure, transport, occupation and domestic activities). Such information is obtained through

standardized questionnaire validated for the Brazilian population [20, 21]. Physically inactive were subjects who did not practice any free-time physical activity in the last three months of the interview and those who did not make intense physical efforts at work, did not commute to work or school and were not responsible for heavy cleaning of the house, as recommended in literature (e.g., they are those who have energy expenditure < 600 METS minutes/week) [22]. For these estimates, sampling weight and sampling strata were considered.

### Data analysis

To estimate hospitalizations and ambulatory costs for breast cancer due to physical inactivity, the Population Attributable Fraction (PAF) calculation strategy was used, which is also used in studies on the global burden of diseases attributable to risk factors [2, 3] and in other studies that estimate the cost of physical inactivity [7–9]. This metric identifies the percentage reduction in the disease, in a given year, if the risk factor (physical inactivity) was not present. That is, if the population met the guidelines for physical activity [22]. For this, the following equation was used: PAF = {[p * (RR -1)] / [p * (RR—1) + 1]}, where 'p' refers to the prevalence of exposure, 'RR' refers to the relative risk attributed to the exposure. RR of the relationship between physical activity and breast cancer was obtained from meta-analysis that analyzed the dose-response between these variables with the inclusion of adjustment covariates based on studies with good methodological quality [23]. Kyu et al. [23] reported that women who practiced physical activity in an amount ≥ 600 MET minutes/week of total physical activity had at least a reduced risk of breast cancer by 3% compared to those who reported lower amounts of physical activity (< 600 MET minutes/week). In the present study RR considered was 1.03, which indicated the risk of breast cancer in women who did not meet physical activity recommendations of ≥ 600 METs minutes/week [22].

Information on the breast cancer was presented in absolute terms and at standardized rates by age. For the calculation of rates, the most current data from estimates of the female population aged ≥ 20 years in each of the Brazilian states and Federal District in 2015, 2016 and 2017 [24] were used. Rates were presented for a population of 100,000 inhabitants. The reference population for the calculation of standardized rates was the estimates of the Ministry of Health of Brazil according to Brazilian capitals and Federal District from 2000 to 2020 [24]. The information on the breast cancer incidence and number and age-standardized rate of hospitalizations (per 100,000 inhabitants) due to breast cancer in Brazilian women are in the S2 and S3 Tables.

All maps were built using TABNET® software (public domain software developed by the Ministry of Health of Brazil) and all images are in the public domain. All cost analyzes were performed using Microsoft Windows Excel® software (Redmond, USA).

## Results

The prevalence of physical inactivity in 2015 among Brazilian women aged ≥ 20 years was 19.5%, 18.5% in 2016, and 19.0% in 2017. Table 1 shows this information according to each Brazilian state capital and Federal District.

In women aged ≥ 20 years (Fig 1), 0.56% of hospitalizations by breast cancer were attributable to physical inactivity. The highest PAF values were found in Brazilian states capitals of Northeastern region, indicating that such locations would prevent greater number of hospitalizations by breast cancer if the population practiced physical activity regularly.

The total hospitalization cost by breast cancer in women aged ≥ 20 years in Brazil from 2015 to 2017 was US$ 33,484,920.54. Of this total, US$ 182,736.76 was due to physical

**Table 1. Prevalence of physical inactivity in women aged ≥ 20 years in 2015, 2016 and 2017, in Brazilian capitals and Federal District.**

| | Prevalence of physical inactivity | | | | | |
| --- | --- | --- | --- | --- | --- | --- |
| | 2015 | | 2016 | | 2017 | |
| | n | % | n | % | n | % |
| Brazil | 6,436 | 19.5 | 5,962 | 18.5 | 6,245 | 19.0 |
| Aracaju | 220 | 17.9 | 243 | 20.0 | 218 | 16.9 |
| Belém | 189 | 16.5 | 238 | 19.4 | 212 | 18.5 |
| Belo Horizonte | 180 | 15.0 | 184 | 14.9 | 182 | 15.5 |
| Boa Vista | 192 | 16.3 | 146 | 14.5 | 139 | 14.6 |
| Brasília | 280 | 23.3 | 199 | 17.2 | 235 | 20.3 |
| Campo Grande | 207 | 17.3 | 233 | 18.9 | 299 | 22.9 |
| Cuiabá | 261 | 19.6 | 189 | 15.4 | 243 | 18.6 |
| Curitiba | 169 | 14.7 | 240 | 19.0 | 266 | 19.1 |
| Florianópolis | 298 | 23.3 | 242 | 20.3 | 291 | 22.8 |
| Fortaleza | 250 | 20.2 | 248 | 20.3 | 257 | 20.2 |
| Goiânia | 281 | 20.9 | 190 | 15.5 | 254 | 19.2 |
| João Pessoa | 344 | 27.2 | 292 | 23.2 | 295 | 23.0 |
| Macapá | 239 | 20.2 | 201 | 18.2 | 146 | 17.4 |
| Maceió | 251 | 20.5 | 266 | 21.3 | 307 | 23.0 |
| Manaus | 213 | 17.6 | 211 | 18.3 | 164 | 16.4 |
| Natal | 265 | 21.3 | 275 | 21.9 | 276 | 22.0 |
| Palmas | 155 | 14.4 | 161 | 15.1 | 158 | 13.7 |
| Porto Alegre | 243 | 20.1 | 241 | 19.1 | 269 | 20.0 |
| Porto Velho | 189 | 16.1 | 176 | 16.2 | 142 | 13.5 |
| Recife | 275 | 21.7 | 284 | 22.0 | 286 | 21.5 |
| Rio Branco | 273 | 21.6 | 154 | 15.5 | 198 | 18.0 |
| Rio de Janeiro | 247 | 19.6 | 246 | 20.7 | 267 | 22.4 |
| Salvador | 243 | 19.2 | 215 | 18.0 | 222 | 17.3 |
| São Luís | 246 | 19.6 | 216 | 18.7 | 244 | 19.1 |
| São Paulo | 190 | 16.2 | 193 | 15.8 | 164 | 13.1 |
| Teresina | 229 | 19.1 | 233 | 19.6 | 247 | 19.8 |
| Vitória | 307 | 23.6 | 246 | 18.8 | 264 | 20.1 |

inactivity. The cities of São Paulo (US$ 35,299.72) and Rio de Janeiro (US$ 22,312.10) were those with the highest hospitalization costs by breast cancer due to physical inactivity (Table 2).

The total ambulatory cost by breast cancer in women aged ≥ 20 years in Brazil from 2015 to 2017 was US$ 207,993,744.39. Of this total, US$ 1,178,841.86 was due to physical inactivity. The cities of São Paulo (US$ 188,863.22) and Rio de Janeiro (US$ 108,470.77) were those with the highest ambulatory costs by breast cancer due to physical inactivity (Table 3).

The mean hospitalization cost for breast cancer due to physical inactivity in the period from 2015 to 2017 among women aged ≥ 20 years was US$ 69,912.25 (± 2,027.11) and this cost was higher in the Brazilian states of Southeastern, Southern and Northeastern regions (Fig 2).

The mean ambulatory cost for breast cancer due to physical inactivity in the period from 2015 to 2017 among women aged ≥ 20 years was US$ 386,201.63 (± 12,553.33) and this cost was higher in the Brazilian states of Southeastern, Southern and Northeastern regions (Fig 3).

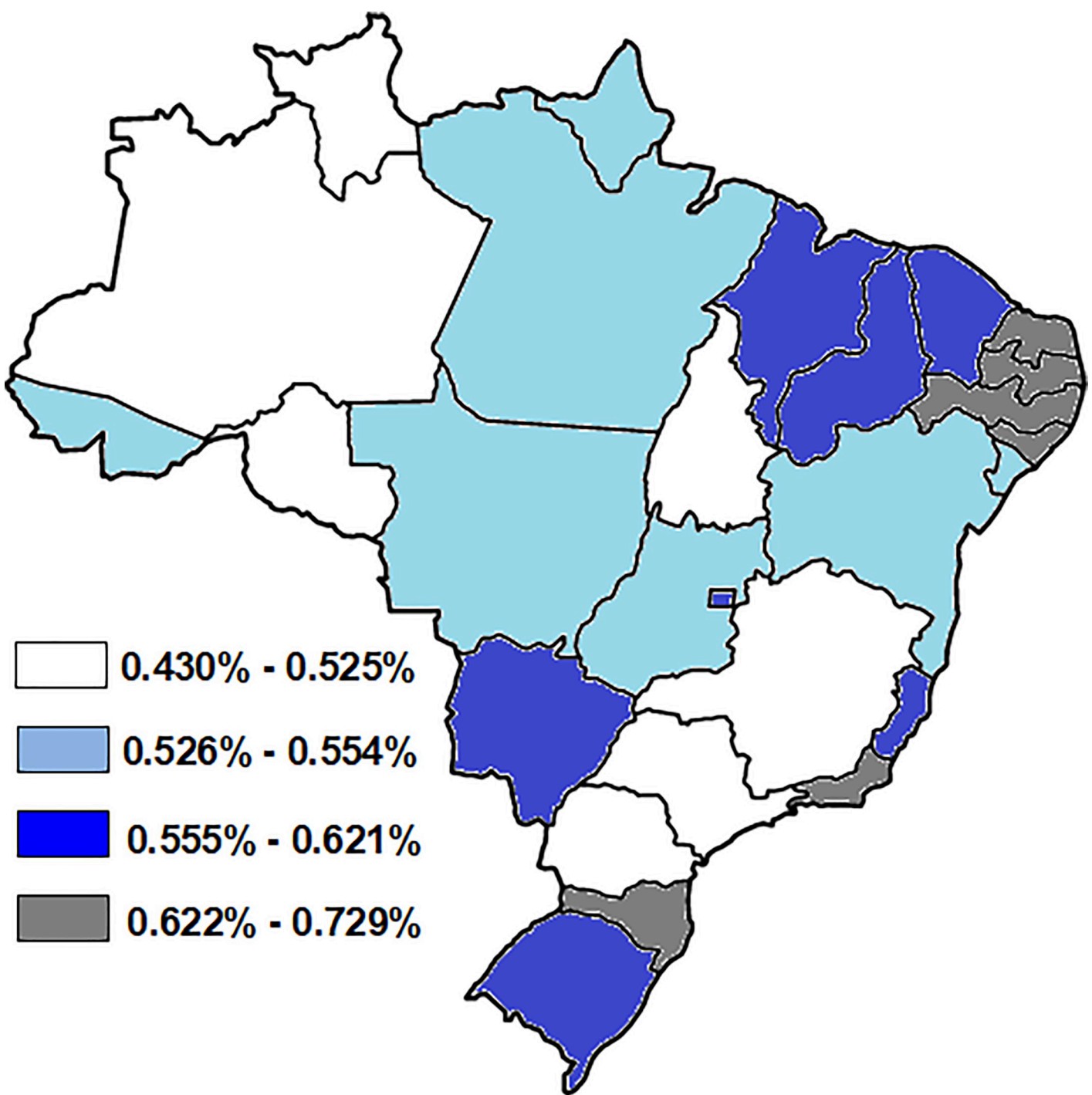

**Fig 1. Population attributable fraction indicating the percentage of hospitalizations for breast cancer due to physical inactivity in women aged ≥ 20 years (Fig 1) from the Brazilian states capitals and Federal District.** All maps were built using TABNET® software (public domain software developed by the Ministry of Health of Brazil) and all images are in the public domain.

## Discussion

The economic burden of physical inactivity was studied by authors from different countries who found high costs for health systems due to physical inactivity [25–27]. Most of these

**Table 2. Cost of hospitalizations for breast cancer in women aged ≥ 20 years in Brazil and in the Brazilian state capitals in 2015, 2016 and 2017.**

| US$ | Total cost of hospitalizations by breast cancer (US$) | | | | Total cost of hospitalizations by breast cancer due to physical inactivity (US$) | | | |
|---|---|---|---|---|---|---|---|---|
| | **2015** | **2016** | **2017** | **Total** | **2015** | **2016** | **2017** | **Total** |
| Brazil | 10,687,409.88 | 11,160,469.79 | 11,637,040.87 | 33,484,920.54 | 59,139.83 | 60,474.44 | 63,122.49 | 182,736.76 |
| Aracaju | 24,433.87 | 41,704.48 | 51,028.93 | 117,167.27 | 130.51 | 248.73 | 257.41 | 636.66 |
| Belém | 137,356.49 | 145,082.04 | 171,264.34 | 453,702.87 | 676.57 | 839.49 | 945.27 | 2,461.33 |
| Belo Horizonte | 723,850.97 | 827,838.24 | 949,217.32 | 2,500,906.54 | 3,242.74 | 3,683.97 | 4,393.43 | 11,320.14 |
| Boa Vista | 17,373.53 | 22,365.80 | 17,984.40 | 57,723.73 | 84.54 | 96.87 | 78.43 | 259.84 |
| Brasília | 269,751.76 | 317,052.65 | 351,963.39 | 938,767.80 | 1,872.48 | 1,627.59 | 2,130.48 | 5,630.55 |
| Campo Grande | 223,031.22 | 290,247.48 | 275,857.40 | 789,136.11 | 1,151.56 | 1,636.42 | 1,882.21 | 4,670.19 |
| Cuiabá | 105,468.38 | 113,445.33 | 102,924.48 | 321,838.20 | 616.53 | 521.71 | 571.13 | 1,709.37 |
| Curitiba | 376,940.91 | 492,409.52 | 571,581.44 | 1,440,931.87 | 1,655.01 | 2,790.83 | 3,256.50 | 7,702.34 |
| Florianópolis | 63,258.25 | 77,592.75 | 55,494.16 | 196,345.16 | 439.11 | 469.68 | 377.00 | 1,285.79 |
| Fortaleza | 740,583.44 | 711,142.95 | 790,174.54 | 2,241,900.93 | 4,460.90 | 4,304.65 | 4,759.61 | 13,525.16 |
| Goiânia | 286,867.08 | 425,047.94 | 453,219.55 | 1,165,134.58 | 1,787.45 | 1,967.32 | 2,595.59 | 6,350.37 |
| João Pessoa | 189,502.17 | 184,655.63 | 261,043.64 | 635,201.44 | 1,533.82 | 1,276.32 | 1,788.86 | 4,599.00 |
| Macapá | 9,766.56 | 11,210.53 | 19,530.33 | 40,507.42 | 58.83 | 60.88 | 101.42 | 221.12 |
| Maceió | 153,443.99 | 158,188.56 | 272,727.25 | 584,359.80 | 937.91 | 1,004.41 | 1,868.92 | 3,811.24 |
| Manaus | 127,671.93 | 155,408.36 | 193,623.29 | 476,703.59 | 670.57 | 848.53 | 947.96 | 2,467.06 |
| Natal | 285,511.46 | 224,746.88 | 278,327.82 | 788,586.16 | 1,812.83 | 1,466.95 | 1,824.92 | 5,104.70 |
| Palmas | 30,588.30 | 25,251.58 | 27,157.71 | 82,997.59 | 131.57 | 113.87 | 111.16 | 356.61 |
| Porto Alegre | 421,819.56 | 416,269.32 | 462,820.66 | 1,300,909.54 | 2,528.33 | 2,371.63 | 2,760.36 | 7,660.32 |
| Porto Velho | 91,802.98 | 81,444.69 | 77,610.60 | 250,858.27 | 441.28 | 393.91 | 313.06 | 1,148.24 |
| Recife | 470,599.54 | 554,286.63 | 569,218.83 | 1,594,105.00 | 3,043.79 | 3,634.31 | 3,647.93 | 10,326.03 |
| Rio Branco | 17,565.87 | 17,162.55 | 28,558.37 | 63,286.79 | 113.09 | 79.44 | 153.39 | 345.92 |
| Rio de Janeiro | 1,169,287.67 | 1,249,706.18 | 1,163,136.18 | 3,582,130.03 | 6,835.22 | 7,712.78 | 7,764.10 | 22,312.10 |
| Salvador | 1,366,024.00 | 1,394,854.46 | 1,256,897.95 | 4,017,776.41 | 7,823.24 | 7,491.76 | 6,489.62 | 21,804.61 |
| São Luís | 276,954.14 | 319,646.21 | 391,882.16 | 988,482.51 | 1,618.97 | 1,783.21 | 2,232.69 | 5,634.87 |
| São Paulo | 2,788,745.33 | 2,566,806.08 | 2,478,585.57 | 7,834,136.97 | 13,487.75 | 12,109.26 | 9,702.71 | 35,299.72 |
| Teresina | 194,228.08 | 212,597.46 | 232,800.02 | 639,625.56 | 1,106.59 | 1,242.77 | 1,374.67 | 3,724.02 |
| Vitória | 124,982.35 | 124,305.50 | 132,410.56 | 381,698.41 | 878.65 | 697.15 | 793.65 | 2,369.46 |

Note. US$: US American Dollar; US$ 1.00 = R$ 0.30 (R$ = Brazilian currency; mean of Brazilian currency from the values of 2015, 2016 and 2017).

studies associated physical inactivity to costs due to cardiovascular disease or diabetes [25–27] and few estimated costs due to breast cancer [7, 9]. The few studies that associated costs to breast cancer were focused on leisure-time physical inactivity [7] and not with physical inactivity in all domains (leisure, transport, domestic activities, work), as was the case in the present study. The estimate of physical inactivity in all domains is important because depending on contextual and social characteristics, the population can be physically active in one domain and not in another [28, 29]. In this sense, estimates of physical inactivity in only one domain may overestimate the economic burden of physical inactivity due to a given chronic disease.

The present study found that from 2015 to 2017, US$ 182,736.76 (annual mean value of US$ 69,912.25) were spent on hospitalizations for breast cancer due to physical inactivity in women aged ≥ 20 years. The ambulatory cost for breast cancer due to physical inactivity from 2015 to 2017 was US$ 1,178,841.86 (average annual value of US$ 386,201.63). These values reflected a total cost of more than US$ 1,361,578.62 in three years due to physical inactivity in Brazil. The Brazilian government developed the "Academia da Saúde" Program, which is a

**Table 3. Ambulatory costs by breast cancer in women aged ≥ 20 years in Brazil and in the Brazilian state capitals in 2015, 2016 and 2017.**

| | Total ambulatory costs by breast cancer (US$) | | | | Total ambulatory costs by breast cancer due to physical inactivity (US$) | | | |
|---|---|---|---|---|---|---|---|---|
| | 2015 | 2016 | 2017 | Total | 2015 | 2016 | 2017 | Total |
| Brazil | 66,547,706.20 | 69,002,663.12 | 72,443,375.07 | 207,993,744.39 | 387,039.90 | 380,851.06 | 410,586.89 | 1,178,841.86 |
| Aracaju | 731,229.11 | 736,308.54 | 615,406.45 | 2,082,944.10 | 3,905.73 | 4,391.50 | 3,104.37 | 11,352.02 |
| Belém | 1,248,804.85 | 1,319,497.61 | 1,544,688.94 | 4,112,991.40 | 6,151.14 | 7,635.04 | 8,525.71 | 22,253.08 |
| Belo Horizonte | 4,972,864.31 | 4,921,002.28 | 5,010,042.61 | 14,903,909.20 | 22,277.64 | 21,898.99 | 23,188.87 | 67,357.85 |
| Boa Vista | 67,924.96 | 22,979.44 | 34,476.93 | 125,381.33 | 330.54 | 99.53 | 150.35 | 566.65 |
| Brasília | 1,078,406.68 | 1,086,120.73 | 1,606,007.16 | 3,770,534.57 | 7,485.74 | 5,575.61 | 9,721.38 | 22,784.24 |
| Campo Grande | 859,786.03 | 937,151.23 | 940,776.37 | 2,737,713.63 | 4,439.25 | 5,283.69 | 6,419.03 | 16,083.48 |
| Cuiabá | 892,048.62 | 980,317.14 | 1,016,485.73 | 2,888,851.49 | 5,214.58 | 4,508.24 | 5,640.52 | 15,400.87 |
| Curitiba | 2,674,829.92 | 2,908,246.71 | 3,041,053.24 | 8,624,129.87 | 11,744.21 | 16,483.05 | 17,325.96 | 45,293.03 |
| Florianópolis | 1,253,623.04 | 1,233,554.91 | 1,147,228.79 | 3,634,406.74 | 8,702.00 | 7,466.88 | 7,793.74 | 23,972.73 |
| Fortaleza | 5,200,153.99 | 5,587,967.46 | 6,134,968.34 | 16,923,089.79 | 31,323.12 | 33,824.73 | 36,953.97 | 102,103.38 |
| Goiânia | 2,151,199.73 | 2,298,807.51 | 2,377,009.61 | 6,827,016.84 | 13,403.98 | 10,639.98 | 13,613.16 | 37,745.26 |
| João Pessoa | 1,805,978.56 | 2,005,025.13 | 2,157,376.76 | 5,968,380.46 | 14,617.51 | 13,858.52 | 14,783.89 | 43,486.74 |
| Macapá | 111,748.00 | 142,659.86 | 178,204.74 | 432,612.61 | 673.11 | 774.69 | 925.40 | 2,400.53 |
| Maceió | 1,407,712.73 | 1,485,944.45 | 1,693,152.97 | 4,586,810.15 | 8,604.52 | 9,434.90 | 11,602.70 | 29,530.73 |
| Manaus | 891,367.71 | 1,082,569.23 | 1,176,336.00 | 3,150,272.94 | 4,681.70 | 5,910.85 | 5,759.24 | 16,390.03 |
| Natal | 2,106,787.06 | 2,046,843.62 | 2,154,155.01 | 6,307,785.69 | 13,376.89 | 13,359.99 | 14,124.20 | 40,860.30 |
| Palmas | 266,723.25 | 253,058.84 | 254,619.66 | 774,401.76 | 1,147.29 | 1,141.19 | 1,042.20 | 3,331.00 |
| Porto Alegre | 4,043,633.64 | 4,123,004.19 | 4,037,748.02 | 12,204,385.85 | 24,236.96 | 23,490.22 | 24,082.00 | 71,824.54 |
| Porto Velho | 152,735.42 | 132,343.52 | 171,541.64 | 456,620.58 | 734.17 | 640.08 | 691.94 | 2,081.72 |
| Recife | 4,887,193.05 | 5,372,720.77 | 5,862,159.59 | 16,122,073.42 | 31,609.85 | 35,227.46 | 37,568.61 | 104,434.94 |
| Rio Branco | 197,607.49 | 185,564.56 | 224,504.01 | 607,676.06 | 1,272.25 | 858.88 | 1,205.81 | 3,329.61 |
| Rio de Janeiro | 5,423,314.09 | 5,930,128.16 | 6,055,319.38 | 17,408,761.63 | 31,702.68 | 36,598.82 | 40,420.12 | 108,470.77 |
| Salvador | 4,523,782.30 | 4,357,461.56 | 4,918,953.59 | 13,800,197.45 | 25,907.76 | 23,403.91 | 25,397.56 | 74,802.65 |
| São Luís | 1,644,602.43 | 1,907,067.35 | 2,051,813.91 | 5,603,483.70 | 9,613.73 | 10,638.96 | 11,689.91 | 31,980.36 |
| São Paulo | 14,111,844.97 | 13,831,915.51 | 14,123,226.21 | 42,066,986.69 | 68,251.86 | 65,253.98 | 55,287.00 | 188,863.22 |
| Teresina | 1,541,923.70 | 1,807,941.17 | 1,696,560.69 | 5,046,425.56 | 8,784.89 | 10,568.55 | 10,018.06 | 29,349.85 |
| Vitória | 2,299,880.55 | 2,306,461.65 | 2,219,558.71 | 6,825,900.90 | 16,168.68 | 12,935.49 | 13,303.72 | 42,394.42 |

Note. US$: US American Dollar; US$ 1.00 = R$ 0.30 (R$ = Brazilian currency; mean of Brazilian currency from the values of 2015, 2016 and 2017).

community program whose objective is to promote physical activity, healthy eating, health education, contributing to the production of care and healthy and sustainable ways of life for the population. To this end, the Program promotes the implementation of public spaces with infrastructure, equipment and qualified professionals [30]. The practice of physical activity is one of the main actions of the program and the entire Brazilian population living in cities where the program has been implemented can participate free of charge [30]. The cost for the implementation of this program varies from R$ 80,000.00 (US$ 15,822.49) to R$ 180.000,00 (US$ 35,600.60) and has the capacity to serve the entire population of cities [30]. That is, the cost for cities to implement community-based programs to promote physical activity is lower than the hospitalization cost for breast cancer due to physical inactivity shown in the present study.

The discussion on the costs of physical inactivity must be done in different countries and about the different contexts of physical activity. The development of programs to promote physical activity in workplaces can also be effective in increasing the levels of physical activity

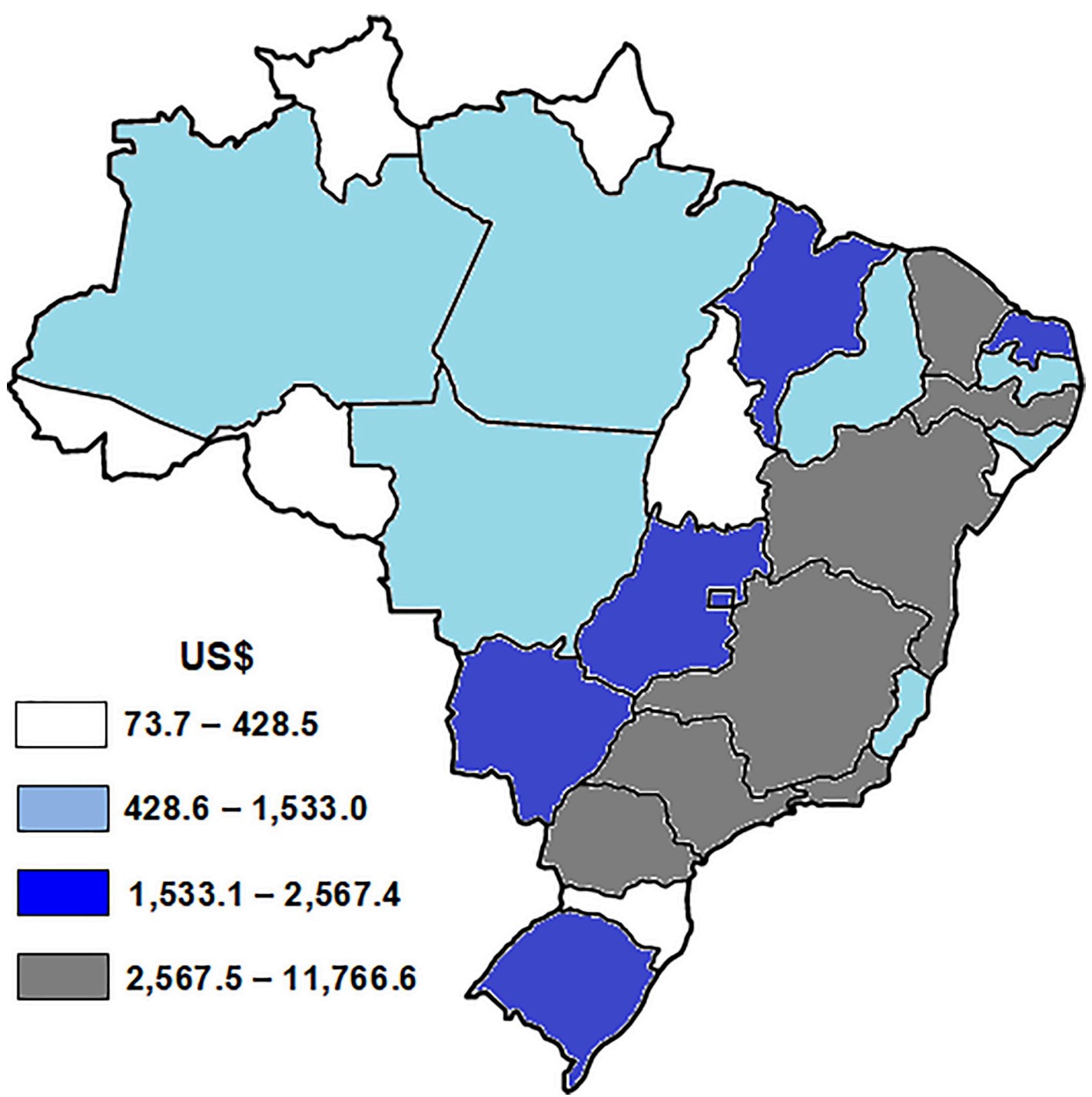

**Fig 2. Average cost of hospitalizations for breast cancer due to physical inactivity in the period of 2015, 2016 and 2017 in the Brazilian states capitals in women aged ≥ 20 years.** All maps were built using TABNET® software (public domain software developed by the Ministry of Health of Brazil) and all images are in the public domain. US$: US American Dollar; US$ 1.00 = R$ 0.30 (R$ = Brazilian currency; mean of Brazilian currency from the values of 2015, 2016 and 2017).

in the population, especially in the adult population that spends most of the day at work [31] and thus can be a strategy for governments to lower the costs of physical inactivity. Lutz et al. [31] developed a systematic review of programs aimed at promoting physical activity in the

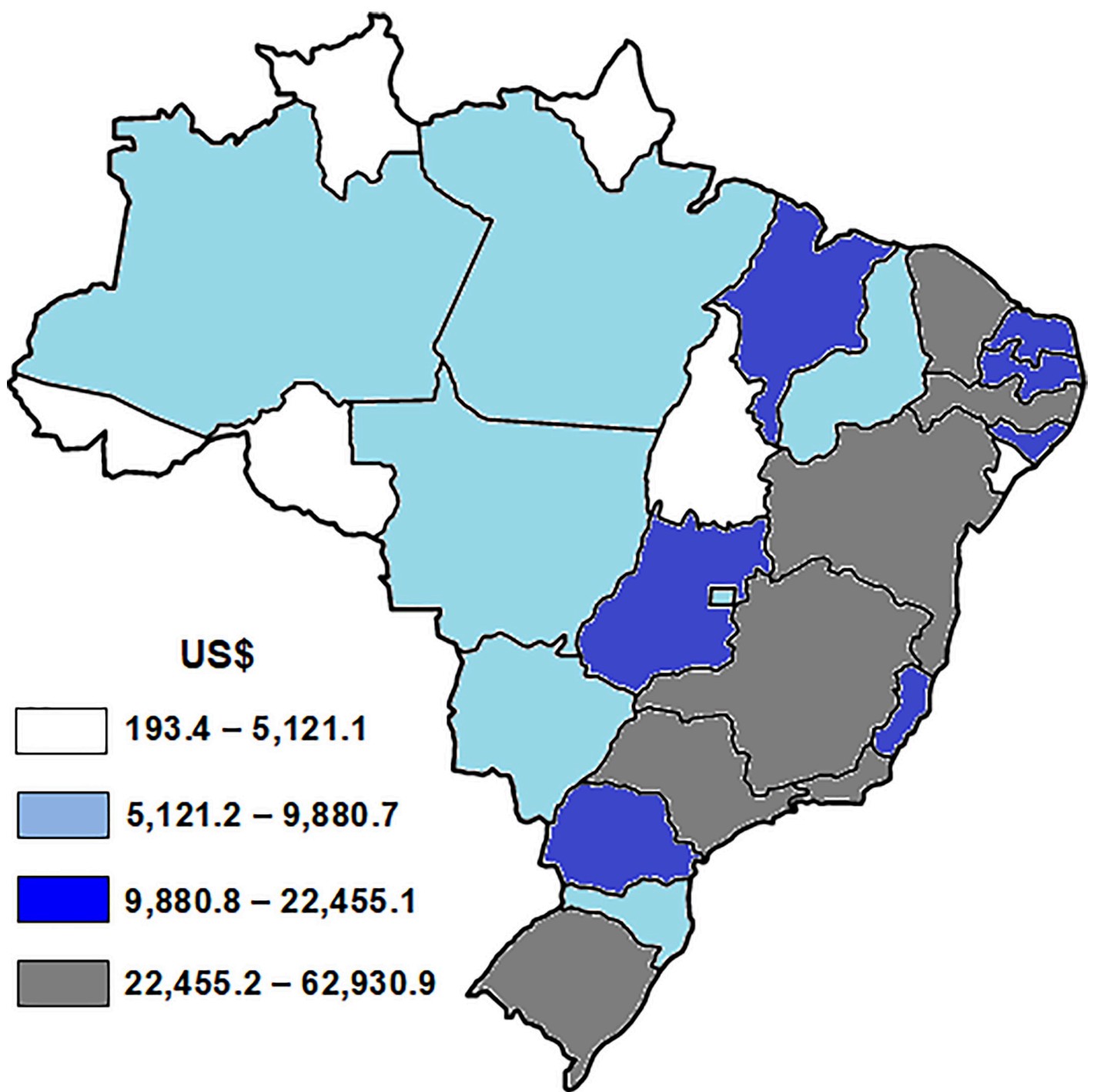

**Fig 3. Average ambulatory cost for breast cancer due to physical inactivity in the period of 2015, 2016 and 2017 in the Brazilian states capitals in women aged ≥ 20 years.** All maps were built using TABNET® software (public domain software developed by the Ministry of Health of Brazil) and all images are in the public domain. US$: US American Dollar; US$ 1.00 = R$ 0.30 (R$ = Brazilian currency; mean of Brazilian currency from the values of 2015, 2016 and 2017).

workplace and estimated the costs of these programs. Of the 16 studies that estimated the costs of implementing this type of program, an average cost per person of € 174.00 was reported, which is equivalent to approximately US$ 206.40 (€ 1.00 = US$ 1.19 in June 18, 2021). In other

words, with estimated cost of US$ 206.40 per person in the work environment, physical activity promotion programs can be implemented, which will prevent breast cancer and other non-communicable diseases and conditions and will consequently reduce the economic burden of hospitalizations due to physical inactivity.

The present study found that in Brazilian women aged $\geq$ 20 years, 0.56% of hospitalizations for breast cancer could be avoided with the regular practice of physical activities. These results reinforce the beneficial effect of the practice of physical activity for the prevention of breast cancer in all age groups, mainly because the practice of physical activity causes changes in female sex hormones (estrogens and progesterone) and in body fat, factors that are related to higher risk of breast cancer [32].

The Brazilian capitals of Northeastern region showed the highest PAF values, indicating that such locations would prevent greater number of hospitalizations by breast cancer if the population practiced physical activity on a regular basis. This information is useful in terms of public health because these capitals are those with the worst economic and social indicators, which also reflects in worse levels of quality and access to health services [3]. In addition, all the Brazilian capitals could save inpatient and outpatient resources if the population were physically active. Thus, increasing the level of physical activity of the population across the country would have a beneficial effect in reducing hospitalizations by breast cancer, which would bring important economic results to these locations, also preventing women from being hospitalized.

The present study developed an analysis of the hospitalization and ambulatory cost by breast cancer. This analysis is one of the possible analyzes for estimating the economic burden of a disease [7]. The cost analysis can only be estimated from variables with measurable values and, therefore, can be accounted for, as is the case with hospitalization. However, regular physical activity can provide numerous health benefits, such as its effects on the reduction in the use of medications [33], and in recovering from some illnesses [34]. Therefore, higher levels of physical activity in the population promote positive effects on economy and health.

There is a lot of debate in the literature about the effect of physical activity on the prevention of breast cancer before and after menopause [35–37]. All health agencies reinforce that the effect of physical activity in preventing breast cancer is more evident after menopause. However, before menopause there is a theoretical discussion that still needs consensus. The World Cancer Research Fund report describes that the strongest evidence that exists in the relationship between physical activity and breast cancer prevention is for women after menopause [35]. Physical activity guidelines describe that there is strong evidence that physical activity prevents breast cancer before and after menopause [36, 37]. The present study adopted the recommendations of the physical activity guidelines because they were the most recent on the subject. In addition, in the present study was used a reference for PAF that which investigated adult and older adults women [23].

This study has several limitations, such as the estimation of hospitalization and ambulatory costs only for breast cancer. Physical inactivity is associated with other health problems that were not investigated in the present study [23], that is, the cost of physical inactivity for the health sector is significantly higher than that estimated in this research. Thus, the results of this study on the economic burden of physical inactivity are much more worrying. Another limitation of this study is the non-stratification of hospitalizations by types of breast cancer and the non-exclusion of hospitalizations due to genetic mutations, as there are types of breast cancer that are more related to lifestyle than others [38]. This study developed analyses at level of Brazilian capitals, and not at level of states. The strategy used in the present study limits the identification of which states or other cities of the Brazil have high hospitalization and ambulatory costs due to physical inactivity and which would need to invest more in physical activity

programs in primary care. The decision to analyze the capitals of Brazil and the Federal District was made because the survey of physical activity that is carried out in Brazil annually is carried out in Brazilian capitals and in the Federal District [17–19]. In addition, another limitation is the temporal difference between exposure to the risk factor and the outcome. We used a RR to estimate the attributable fraction from a systematic review that led to studies conducted in developed countries and this is another limitation. The study of ecological design is not free from the ecological fallacy. Finally, estimates of physical inactivity in the present study came from surveys that used self-reported physical activity measures, which are less accurate than objective measures [39].

## Conclusions

It could be concluded that physical inactivity has contributed to high number of hospitalizations by breast cancer in Brazilian female population, which resulted in cost of more than US$ 182,700.00 from 2015 to 2017 for health services. In addition, the ambulatory cost for breast cancer due to physical inactivity from 2015 to 2017 was US$ 1,178,841.86. These values reflected a total cost of more than US$ 1,300,000.00 in three years due to physical inactivity in Brazil. Thus, the promotion of physical activity in the Brazilian female population would bring economic benefits for all geographic regions.

## Supporting information

**S1 Table. Description of outpatient procedures on breast cancer applied in Brazil.** DATA-SUS (2015, 2016 and 2017).
(DOC)

**S2 Table. Incidence of breast cancer among women in Brazilian state capitals and Federal District (aged ≥ 20 years).** *rate per 100,000 inhabitants. Reference population: Reference population: women residing in each capital for each year (aged ≥ 20 years).
(DOC)

**S3 Table. Number and age-standardized rate of hospitalizations (per 100,000 inhabitants) due to breast cancer in women aged ≥ 20 years in 2015, 2016 and 2017, in Brazil and in the Brazilian state capitals.** *Age-standardized (per 100,000 inhabitants).
(DOC)

## Author Contributions

**Conceptualization:** Diego Augusto Santos Silva.

**Formal analysis:** Diego Augusto Santos Silva.

**Methodology:** Diego Augusto Santos Silva.

**Project administration:** Diego Augusto Santos Silva.

**Writing – original draft:** Diego Augusto Santos Silva.

**Writing – review & editing:** Diego Augusto Santos Silva.

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
