## [Decision Letter · Decision Letter 0]

7 Jun 2021

PONE-D-21-09895

The economic burden of breast cancer due to physical inactivity in Brazil

PLOS ONE

Dear,

Thank you for submitting your manuscript to PLOS ONE. After careful consideration, we feel that it has merit but does not fully meet PLOS ONE’s publication criteria as it currently stands. Therefore, we invite you to submit a revised version of the manuscript that addresses the points raised during the review process.

 Please correct the manuscript according to the comment given by reviewer.

We look forward to receiving your revised manuscript.

Kind regards,

Muhammad Shahzad Aslam, Ph.D.,M.Phil., Pharm-D

Academic Editor

PLOS ONE

Journal Requirements:

2.We note that Figure(s) 1 and 2 in your submission contain map images which may be copyrighted. All PLOS content is published under the Creative Commons Attribution License (CC BY 4.0), which means that the manuscript, images, and Supporting Information files will be freely available online, and any third party is permitted to access, download, copy, distribute, and use these materials in any way, even commercially, with proper attribution. For these reasons, we cannot publish previously copyrighted maps or satellite images created using proprietary data, such as Google software (Google Maps, Street View, and Earth). For more information, see our copyright guidelines: http://journals.plos.org/plosone/s/licenses-and-copyright.

a)  You may seek permission from the original copyright holder of Figure(s) 1 and 2 to publish the content specifically under the CC BY 4.0 license. 

Reviewers' comments:

Reviewer's Responses to Questions

**Comments to the Author**

1. Is the manuscript technically sound, and do the data support the conclusions?

Reviewer #1: Yes

Reviewer #2: Yes

Reviewer #3: Partly

Reviewer #4: Partly

Reviewer #5: Yes

Reviewer #6: Partly

2. Has the statistical analysis been performed appropriately and rigorously? 

Reviewer #1: Yes

Reviewer #2: Yes

Reviewer #3: Yes

Reviewer #4: Yes

Reviewer #5: Yes

Reviewer #6: No

3. Have the authors made all data underlying the findings in their manuscript fully available?

Reviewer #1: Yes

Reviewer #2: Yes

Reviewer #3: Yes

Reviewer #4: Yes

Reviewer #5: Yes

Reviewer #6: Yes

4. Is the manuscript presented in an intelligible fashion and written in standard English?

Reviewer #1: Yes

Reviewer #2: Yes

Reviewer #3: Yes

Reviewer #4: Yes

Reviewer #5: No

Reviewer #6: No

5. Review Comments to the Author

Reviewer #1: The manuscript "The economic burden of breast cancer due to physical inactivity in Brazil" study the economic impact of physical inactivity and breast cancer to Brazilian healthcare system. The study has merit and was written. However, I have some comments to the author.

Major comments:

1) There is only one author in the study? If is true, the sentence “These authors contributed equally to this work” in the first page make no sense.

2) The correlation between mean hospitalization cost and healthcare quality and access index is weak both in women aged 20-59 years and in those aged ≥60 years (rho = 0.37, p = 0.04 and rho = 0.43, p = 0.02) (figure 1 and 2). Besides some Brazilian estates showed a higher cost and higher access index, this is not direct correlated in the country. This should be clear in results and discussion.

3) In the discussion, lines 353 to 355 “The cost for the implementation of this program varies from R$ 80.000,00 (US$ 266,400.00) to R$ 180.000,00 (US$ 599,400.00) and ….”. The values in Real and Dollar should be revised in this sentence and perhaps in all text.

Minor comments:

1) Values of rho and p should be corrected. For instance: rho = .37, p = .04 to rho= 0.37, p = 0.04

Reviewer #2: I found that the article is well written. The content is interesting and well elaborated.

I found same minor errors that should be seen.

LIne 62 probably the number in euros is wrongly written.

Line 185 the term PAF was used in line 251 the term FAP was used for the same meaning

Line 342 is this phrase correct?

would be € 133,074, which is equivalent to approximately US$ 151,000,000

Line 354

implementation of this program varies from R$ 80.000,00 (US$ 266,400.00) to R$ 354 180.000,00 (US$ 599,400.00)

Reviewer #3: The study is important to added the information about the correlation about a breast cancer and economic costs in populous country such as brazil. The Datasus is a main source of data health from Brazilian`s population. This study have not study design, and the authors did not declare the conflict of interest.

Please see attachment.

Reviewer #4: General comments: The article has an exciting and very current theme. Economic analyzes in low- and middle-income countries are always welcome. Three methodological issues are central. The first question refers to the option of the authors to use a survey of the prevalence of risk factors that is not nationally representative to build a fraction attributable to be applied nationally. The second question concerns using only direct hospital expenses and highlighting in the title that an analysis of the economic burden is a broad concept. The third and last concerns the questionable contribution of the correlation analysis between attributable expenditure and the Healthcare Access and Quality (HAQ) Index.

Specific comments: 1. Although the article's title refers to economic burden in Brazil, data from Vigitel were used, which is a telephone survey with representation only for the 26 capitals and DF and not for Brazil; 2. The economic burden seems a broad term. The author only analyzed the hospital expenses of the public health system; 3. It was unclear why choosing the analysis period between 2015 and 2017; 4. It was also unclear why analyzing the correlation between expenditures and conditions of access and quality of services (HAQ) in the states; 4. It seems that the approval of CONEP was from the telephone survey research (Vigitel) and not from the study carried out; 5.What is the purpose of calculating the standardized rate of hospitalization for breast cancer; 6. the value of RR 1.40 used was not found in the reference text; 7. The example used in the discussion (lines 336-344) does not seem suitable for the discussion section once it was a foreign factual experience; 8. Review the dollar amounts in lines 352-355; 9. The reference used in lines 358-370 also does not seem adequate since an analysis of the attributable fraction of expenses by type (recreational, occupational, transport, at home) was not performed. The values marked in the article were not found in the original 10. Clarify the justification pointed out in lines 408-412; 11. The quality of life and mental well-being are considered non-measurable; why? Lines 412-416; 12. In the study's limitations, there was no mention of the issue of temporal GAP between exposure to the risk factor and the outcome. This question has not been considered, although other research groups on attributable fraction consider it very relevant; 13. I could not access reference #38, review doi; 14. Lines 425-426 reinforce that attributable expenditures were carried out using prevalence data from the state's capitals. Wouldn't that be a limitation? 15. No analysis was made of the correlation between expenditure attributable to physical inactivity and the HAQ index. Why? since the rho values are relatively low.

Reviewer #5: The introduction section is so long. Along the stretch between 62 and 81 lines are highlighted some studies with cost estimates widely different, limiting the cross-countries comparison. Sugiro excluir a maior parte e/ou transferir uma parte para a discussão. I suggest deleting most of this and/or transferring a part to the discussion.

It is not clear which is the study design.

The authors argue that “For all Brazilian states, the quality of data from DATASUS is considered high and close to high-income countries”. However, several authors suggest which coverage and quality of the North Brazilian region datasets is poor in regard south region, for example (Please, check this manuscripts: https://pubmed.ncbi.nlm.nih.gov/31800858/;
https://pubmed.ncbi.nlm.nih.gov/31859881/).

Please consider exclude the following stretch “All DATASUS information follows the ethical precepts of the Declaration of Helsinki, so that users' anonymity is respected.”, because the study is based in secondary data.

How the authors dealt with the number of readmissions?

If the quality of data from DATASUS is considered high in Brazil why the authors did not use the breast cancer Brazilian estimates?

The Healthcare Access and Quality (HAQ) Index may have been influenced by the poor-quality data of the north region cities. Are the north region estimates reliable?

Reviewer #6: This study highlights an important topic on the impact of physical inactivity on hospitalization costs due to breast cancer, through secondary analysis of national databases. The study aimed “to estimate the economic burden of breast cancer due to physical inactivity in the Brazilian female population over a three-year period (2015 to 2017) and to relate these costs to conditions of access and quality of health services in Brazilian states”.

I have several major and some minor observations.

1) A first concern is using the VIGITEL database to estimate the prevalence of physical inactivity of Brazilians. This database is restricted to the Brazilian population aged ≥ 18 years living in households served by at least one fixed telephone line in 26 Brazilian capitals and the Federal District. Then, it is not representative of the Brazilian population, primarily because the exposure investigated (physical activity) can differ significantly between female residents of capitals and other cities in the country. The title, abstract, analyses, results, and conclusion should be revised to attend to this particularity.

2) In the Abstract, the author stated that the “Population Attributable Fraction” (PAF) was calculated but did not present the results, an important result of this study.

3) In the Introduction (lines 61-67), it stated some cost estimates of breast cancer in different countries and, importantly, in different points of time (from 2007 to 2020). This difference in time could be influencing in the estimates presented. I suggest considering presenting more current data and standardize the values in dollars.

4) In the Methods (lines 97-112), the databases used in the present study should be presented more clearly. Which data were obtained from DATAUS (SIH?) and which data were obtained from VIGITEL surveys? Are DATASUS and VIGITEL different databases with specific aims?

5) In lines 99-100, the author stated that “the quality of data from DATASUS is considered high and close to high-income countries”, but the references used to support this claim (11,12) deal whit mortality statistics' information, not included in this study. Please review this claim and references.

6) In lines 129-132, it is stated the data extracted from the DATASUS (total number of hospitalizations and the cost of these hospitalizations in the Brazilian female population aged 20-59 years and ≥60 years, according to year). In lines 325-326, the author informs that analyses were conducted at the country's geographic region and states and not at municipality level. It is unclear to me why the analyses were not restricted to 26 Brazilian capitals and the Federal District?

7) The RR used to estimate the attributable fraction was taken from a systematic review (Kyu 2013) (lines 192-195) that led to studies conducted in developed countries with populations different from the Brazilian one (United States, Canada, European countries, Japan and China). This limitation should be pointed out in the discussion.

8) Please consider presenting the results by capitals and Federal District, not by Brazilian states.

9) Lines 299-300. The statistic used (Spearman's correlation coefficient) was not described in the Method.

10) In lines 320-321, it is stated that “few estimated costs due to breast cancer [7]”. Please consider to cite other studies, such as Ding, 2016 (ref 9).

11) In lines 366-370, the author stated that “[…] physical activity promotion programs can be implemented, which will prevent breast cancer and other non-communicable diseases and conditions.” Despite the recognized importance of these programs, they do not guarantee the prevention of all breast cancer cases, given the existence of other risk factors for the disease.

12) It would be great if the author could include a limitation about the susceptibility of the analysis conducted in the present study to ecological fallacy.

13) Please change FAP (lines 251 and 379) by PAF.

6. PLOS authors have the option to publish the peer review history of their article (what does this mean?). If published, this will include your full peer review and any attached files.

Reviewer #1: No

Reviewer #2: **Yes: **João Pedreira Duprat Neto

Reviewer #3: **Yes: **Jonas Baltazar Daniel ( Biologist, MSc Nutrition Food & Health, PhD candidate -Public Health)

Reviewer #4: No

Reviewer #5: No

Reviewer #6: No

---

## [Author Response · Author response to Decision Letter 0]

24 Jun 2021

Reviewer #1:

The manuscript "The economic burden of breast cancer due to physical inactivity in Brazil" study the economic impact of physical inactivity and breast cancer to Brazilian healthcare system. The study has merit and was written. However, I have some comments to the author.

Authors: We really appreciate your comments on the article.

Reviewer #1:

Major comments:

1) There is only one author in the study? If is true, the sentence “These authors contributed equally to this work” in the first page make no sense.

Authors: The article has only one author. In the new version of the article the sentence was corrected to "This author contributed to all the work”.

Reviewer #1:

2) The correlation between mean hospitalization cost and healthcare quality and access index is weak both in women aged 20-59 years and in those aged ≥60 years (rho = 0.37, p = 0.04 and rho = 0.43, p = 0.02) (figure 1 and 2). Besides some Brazilian estates showed a higher cost and higher access index, this is not direct correlated in the country. This should be clear in results and discussion.

Authors: We really appreciate your comments. In the new version of the article we only added information about Brazilian capitals as suggested by other reviewers. These new data demonstrated that there was no relationship between mean hospitalization cost and healthcare quality and access index. In addition, in discussion section we added the information: “The Brazilian states of Northern and Northeastern regions showed the highest PAF values, indicating that such locations would prevent greater number of hospitalizations by breast cancer if the population practiced physical activity on a regular basis. In addition, the present study found that Brazilian capitals with lower values of Healthcare Quality and Access index had a higher percentage of hospitalizations by breast cancer due to physical inactivity. This information is useful in terms of public health because these geographical regions are those with the worst economic and social indicators, which also reflects in worse levels of quality and access to health services [3]. Thus, increasing the level of physical activity of the population in these geographic regions would have a beneficial effect in reducing hospitalizations by breast cancer, which would bring important economic results to these locations, also preventing women from being hospitalized and not having the adequate care and treatment for breast cancer, as these locations are those with poor infrastructure and fewer health professionals to serve the population compared to other Brazilian regions [3]”. 

Reviewer #1:

3) In the discussion, lines 353 to 355 “The cost for the implementation of this program varies from R$ 80.000,00 (US$ 266,400.00) to R$ 180.000,00 (US$ 599,400.00) and ….”. The values in Real and Dollar should be revised in this sentence and perhaps in all text.

Authors: The change was made: “…The cost for the implementation of this program varies from R$ 80,000.00 (US$ 15,822.49) to R$ 180.000,00 (US$ 35,600.60)…”. 

Reviewer #1:

Minor comments:

1) Values of rho and p should be corrected. For instance: rho = .37, p = .04 to rho= 0.37, p = 0.04

Authors: The change was made. 

Reviewer #2: I found that the article is well written. The content is interesting and well elaborated.

I found same minor errors that should be seen.

Authors: We really appreciate your comments on the article.

Reviewer #2: 

Line 62 probably the number in euros is wrongly written.

Authors: The change was made

Reviewer #2: 

Line 185 the term PAF was used in line 251 the term FAP was used for the same meaning

Authors: In the new version of the paper we put all terms as “PAF”. 

Reviewer #2: 

Line 342 is this phrase correct?

would be € 133,074, which is equivalent to approximately US$ 151,000,000

The authors estimated different cost scenarios and reported that the highest estimated cost would be € 133,074.00, which is equivalent to approximately Authors: In the new version of the paper we correct the information: “…The authors estimated different cost scenarios and reported that the highest estimated cost would be € 133,074.00, which is equivalent to approximately US$ 162,040.00 [33] (€ 1.00 = US$ 1.22 in June 10, 2021)…”.

Reviewer #2: 

Line 354

implementation of this program varies from R$ 80.000,00 (US$ 266,400.00) to R$ 354 180.000,00 (US$ 599,400.00)

Authors: In the new version of the paper we correct the information: “…The cost for the implementation of this program varies from R$ 80,000.00 (US$ 15,822.49) to R$ 180.000,00 (US$ 35,600.60) and has the capacity to serve the entire population of cities…”.

Reviewer #3: 

The study is important to added the information about the correlation about a breast cancer and economic costs in populous country such as brazil. The Datasus is a main source of data health from Brazilian`s population. This study have not study design, and the authors did not declare the conflict of interest.

Please see attachment.

Authors: We really appreciate your comments on the article.

Reviewer #4: 

General comments: The article has an exciting and very current theme. Economic analyzes in low- and middle-income countries are always welcome. Three methodological issues are central. The first question refers to the option of the authors to use a survey of the prevalence of risk factors that is not nationally representative to build a fraction attributable to be applied nationally. The second question concerns using only direct hospital expenses and highlighting in the title that an analysis of the economic burden is a broad concept. The third and last concerns the questionable contribution of the correlation analysis between attributable expenditure and the Healthcare Access and Quality (HAQ) Index.

Authors: We really appreciate your comments on the article. We agree that it is a limitation of the article to investigate only Brazilian capitals. This information was added in the discussion section as a limitation: “…This study developed analyses at level of Brazilian states capitals, and not at level of states. The strategy used in the present study limits the identification of which states or other cities of the Brazil have high hospitalization costs due to physical inactivity and which would need to invest more in physical activity programs in primary care...”. 

In the new version of the paper we changed the title and the aim of this study. 

Title: Hospitalization cost related to breast cancer due to physical inactivity in the Brazilian state capitals; Objective: to estimate the hospitalization cost related to breast cancer due to physical inactivity in the Brazilian states capitals over a three-year period (2015 to 2017) and to relate these costs to conditions of access and quality of health services in Brazilian states.

Reviewer #4: 

Specific comments: 1. Although the article's title refers to economic burden in Brazil, data from Vigitel were used, which is a telephone survey with representation only for the 26 capitals and DF and not for Brazil; 

Authors: We really appreciate your comments on the article. We agree that it is a limitation of the article to investigate only Brazilian capitals. Because of that we changed all the results. In addition, this information was added in the discussion section as a limitation: “…This study developed analyses at level of Brazilian states capitals, and not at level of states. The strategy used in the present study limits the identification of which states or other cities of the Brazil have high hospitalization costs due to physical inactivity and which would need to invest more in physical activity programs in primary care...”. 

Reviewer #4: 

2. The economic burden seems a broad term. The author only analyzed the hospital expenses of the public health system; 

Authors: In the new version of the paper we changed the title and the aim of this study. 

Title: Hospitalization cost related to breast cancer due to physical inactivity in the Brazilian state capitals; Objective: to estimate the hospitalization cost related to breast cancer due to physical inactivity in the Brazilian states capitals over a three-year period (2015 to 2017) and to relate these costs to conditions of access and quality of health services in Brazilian states.

Reviewer #4: 

3. It was unclear why choosing the analysis period between 2015 and 2017; 

Authors: We chose these years because they are the years with data consolidated in DATASUS. The most recent data available in DATASUS (2018, 2019 and 2020) are still in the process of consolidation by the national system.

Reviewer #4: 

4. It was also unclear why analyzing the correlation between expenditures and conditions of access and quality of services (HAQ) in the states; 

Authors: In the new version of the article we added new correlations between percentage of hospitalizations for breast cancer due to physical inactivity and the Healthcare Quality and Access index. The information allowed progress in the discussions of the article.

Reviewer #4: 

4. It seems that the approval of CONEP was from the telephone survey research (Vigitel) and not from the study carried out; 

Authors: The data collected directly from people was from VIGITEL, because of that the Ethics Committee was referring to this research. DATASUS data are secondary and free to access.

Reviewer #4: 

5.What is the purpose of calculating the standardized rate of hospitalization for breast cancer; 

Authors: It is important to calculate the standardized rate because the standardized rate takes into account the population of the city. If the rate is not standardized, the information will not be real for the population of interest (Please visit:

https://www.cdc.gov/cancer/uscs/technical_notes/stat_methods/rates.htm#:~:text=Crude%20rates%20are%20influenced%20by,cancers%20increase%20with%20increasing%20age).

Reviewer #4: 

6. the value of RR 1.40 used was not found in the reference text; 

Authors: In the new version of the paper we changed the RR. RR of the relationship between physical activity and breast cancer was obtained from meta-analysis that analyzed the dose-response between these variables with the inclusion of adjustment covariates based on studies with good methodological quality (Kyu et al, 2016). Kyu et al. (2016) reported that women who practiced physical activity in an amount ≥ 600 MET minutes/week of total physical activity had at least a reduced risk of breast cancer by 3% compared to those who reported lower amounts of physical activity. In the present study RR considered was 1.03, which indicated the risk of breast cancer in women who did not meet physical activity recommendations of 600 METs minutes/week.

Reviewer #4: 

7. The example used in the discussion (lines 336-344) does not seem suitable for the discussion section once it was a foreign factual experience; 

Authors: In the new version of the paper we removed the Germany article from the discussion. 

Reviewer #4: 

8. Review the dollar amounts in lines 352-355; 

Authors: The change was made. 

Reviewer #4: 

9. The reference used in lines 358-370 also does not seem adequate since an analysis of the attributable fraction of expenses by type (recreational, occupational, transport, at home) was not performed. The values marked in the article were not found in the original 

Authors: In the new version of the paper we kept the article in the discussion section because it brings a debate about the implementation of physical activity programs in other contexts, which can be a strategy to be used in Brazil. 

The value marked in our research is in the pag. 135 of the paper of Lutz et al. (2020).

- Lutz N, Clarys P, Koenig I, Deliens T, Taeymans J, Verhaeghe N. Health economic evaluations of interventions to increase physical activity and decrease sedentary behavior at the workplace: a systematic review. Scand J Work Environ Health. 2020;46(2):127-42. doi: https://doi.org/10.5271/sjweh.3871

Reviewer #4: 

10. Clarify the justification pointed out in lines 408-412; 11. The quality of life and mental well-being are considered non-measurable; why? Lines 412-416; 

Authors: We really appreciate your comments on the article. In the new version of the paper we changed this paragraph: “...The present study developed an analysis of the hospitalization cost by breast cancer. This analysis is one of the possible analyzes for estimating the economic burden of a disease [7]. The cost analysis can only be estimated from variables with measurable values and, therefore, can be accounted for, as is the case with hospitalization. However, regular physical activity can provide numerous health benefits, such as its effects on the reduction in the use of medications [38], and in recovering from some illnesses [39]. Therefore, higher levels of physical activity in the population promote positive effects on economy and health…”. 

Reviewer #4: 

12. In the study's limitations, there was no mention of the issue of temporal GAP between exposure to the risk factor and the outcome. This question has not been considered, although other research groups on attributable fraction consider it very relevant.

Authors: In the new version of the paper we added this limitation. 

Reviewer #4: 

13. I could not access reference #38, review doi

Authors: The change was made. 

Reviewer #4: 

14. Lines 425-426 reinforce that attributable expenditures were carried out using prevalence data from the state's capitals. Wouldn't that be a limitation?

Authors: In the new version of the paper all the information was from state’s capitals. 

Reviewer #4: 

15. No analysis was made of the correlation between expenditure attributable to physical inactivity and the HAQ index. Why? since the rho values are relatively low.

Authors: In the new version of the article we added new correlations between percentage of hospitalizations for breast cancer due to physical inactivity and the Healthcare Quality and Access index. The information allowed progress in the discussions of the article.

Reviewer #5: 

The introduction section is so long. Along the stretch between 62 and 81 lines are highlighted some studies with cost estimates widely different, limiting the cross-countries comparison. Sugiro excluir a maior parte e/ou transferir uma parte para a discussão. I suggest deleting most of this and/or transferring a part to the discussion.

Authors: The different reviewers of the paper have debated the different sections of the research. We have chosen to keep the introduction information because, along with other reviewers, we understand that it is important for the reader to bring information from previous studies about the costs of physical inactivity.

Reviewer #5: 

It is not clear which is the study design.

Authors: Our paper is an ecological study. We added the information on the method section. 

Reviewer #5: 

The authors argue that “For all Brazilian states, the quality of data from DATASUS is considered high and close to high-income countries”. However, several authors suggest which coverage and quality of the North Brazilian region datasets is poor in regard south region, for example (Please, check this manuscripts: https://pubmed.ncbi.nlm.nih.gov/31800858/;
https://pubmed.ncbi.nlm.nih.gov/31859881/).

Authors: The quality of data from DATASUS is considered high and close to high-income countries. I understand that some information still needs further improvement and it does need to improve. However, DATASUS is a quality system whose data is important for Brazil. 

Reviewer #5: 

Please consider exclude the following stretch “All DATASUS information follows the ethical precepts of the Declaration of Helsinki, so that users' anonymity is respected.”, because the study is based in secondary data.

Authors: The change was made. 

Reviewer #5: 

How the authors dealt with the number of readmissions?

Authors: We do not consider this information in the article.

Reviewer #5: 

If the quality of data from DATASUS is considered high in Brazil why the authors did not use the breast cancer Brazilian estimates?

Authors: In the new version of the paper we added all the information from DATASUS (included breast cancer estimates). “…Information regarding the incidence of breast cancer in the Brazilian female population aged 20-59 years and aged ≥ 60 years over 2015, 2016 and 2017 was taken from estimates of the DATASUS (i.e., Panel-Oncology), which analyzes estimates and makes information available free of charge”.

Reviewer #5: 

The Healthcare Access and Quality (HAQ) Index may have been influenced by the poor-quality data of the north region cities. Are the north region estimates reliable?

Authors: The quality of data from DATASUS is considered high and close to high-income countries. I understand that some information still needs further improvement and it does need to improve. However, DATASUS is a quality system whose data is important for Brazil. 

Reviewer #6: 

This study highlights an important topic on the impact of physical inactivity on hospitalization costs due to breast cancer, through secondary analysis of national databases. The study aimed “to estimate the economic burden of breast cancer due to physical inactivity in the Brazilian female population over a three-year period (2015 to 2017) and to relate these costs to conditions of access and quality of health services in Brazilian states”.

I have several major and some minor observations.

1) A first concern is using the VIGITEL database to estimate the prevalence of physical inactivity of Brazilians. This database is restricted to the Brazilian population aged ≥ 18 years living in households served by at least one fixed telephone line in 26 Brazilian capitals and the Federal District. Then, it is not representative of the Brazilian population, primarily because the exposure investigated (physical activity) can differ significantly between female residents of capitals and other cities in the country. The title, abstract, analyses, results, and conclusion should be revised to attend to this particularity.

Authors: We really appreciate your comments. In the new version of the article we only added information about Brazilian capitals as suggested.

Reviewer #6: 

2) In the Abstract, the author stated that the “Population Attributable Fraction” (PAF) was calculated but did not present the results, an important result of this study.

Authors: The change was made. 

Reviewer #6: 

3) In the Introduction (lines 61-67), it stated some cost estimates of breast cancer in different countries and, importantly, in different points of time (from 2007 to 2020). This difference in time could be influencing in the estimates presented. I suggest considering presenting more current data and standardize the values in dollars.

Authors: The change was made for all information. 

Reviewer #6: 

4) In the Methods (lines 97-112), the databases used in the present study should be presented more clearly. Which data were obtained from DATAUS (SIH?) and which data were obtained from VIGITEL surveys? Are DATASUS and VIGITEL different databases with specific aims?

Authors: We used the information of the SIH for hospitalization due to breast cancer. The information about physical activity were from Vigitel. In the new version of the paper we added the information. 

Reviewer #6: 

5) In lines 99-100, the author stated that “the quality of data from DATASUS is considered high and close to high-income countries”, but the references used to support this claim (11,12) deal whit mortality statistics' information, not included in this study. Please review this claim and references.

Authors: The mortality statistics' information are from DATASUS. Because of that we kept the reference.

Reviewer #6: 

6) In lines 129-132, it is stated the data extracted from the DATASUS (total number of hospitalizations and the cost of these hospitalizations in the Brazilian female population aged 20-59 years and ≥60 years, according to year). In lines 325-326, the author informs that analyses were conducted at the country's geographic region and states and not at municipality level. It is unclear to me why the analyses were not restricted to 26 Brazilian capitals and the Federal District?

Authors: We really appreciate your comments. In the new version of the article we only added information about Brazilian capitals. We changed all the results. In the new version of the article we added new correlations between percentage of hospitalizations for breast cancer due to physical inactivity and the Healthcare Quality and Access index. The information allowed progress in the discussions of the article.

Reviewer #6: 

7) The RR used to estimate the attributable fraction was taken from a systematic review (Kyu 2013) (lines 192-195) that led to studies conducted in developed countries with populations different from the Brazilian one (United States, Canada, European countries, Japan and China). This limitation should be pointed out in the discussion.

Authors: We really appreciate your comments. We added this limitation. 

Reviewer #6: 

8) Please consider presenting the results by capitals and Federal District, not by Brazilian states.

Authors: In the new version of the article we only added information about Brazilian capitals. We changed all the results. In the new version of the article we added new correlations between percentage of hospitalizations for breast cancer due to physical inactivity and the Healthcare Quality and Access index. The information allowed progress in the discussions of the article.

Reviewer #6: 

9) Lines 299-300. The statistic used (Spearman's correlation coefficient) was not described in the Method.

Authors: In the new version of the article we added the information. 

Reviewer #6: 

10) In lines 320-321, it is stated that “few estimated costs due to breast cancer [7]”. Please consider to cite other studies, such as Ding, 2016 (ref 9).

Authors: The changed was made. 

Reviewer #6: 

11) In lines 366-370, the author stated that “[…] physical activity promotion programs can be implemented, which will prevent breast cancer and other non-communicable diseases and conditions.” Despite the recognized importance of these programs, they do not guarantee the prevention of all breast cancer cases, given the existence of other risk factors for the disease.

Authors: In the new version of the article we added others information. 

Reviewer #6: 

12) It would be great if the author could include a limitation about the susceptibility of the analysis conducted in the present study to ecological fallacy.

Authors: In the new version of the article we added this limitation. 

Reviewer #6: 

13) Please change FAP (lines 251 and 379) by PAF. 

Authors: The changed was made.

---

## [Decision Letter · Decision Letter 1]

13 Jul 2021

PONE-D-21-09895R1

Hospitalization cost related to breast cancer due to physical inactivity in the Brazilian state capitals

PLOS ONE

Dear,

Thank you for submitting your manuscript to PLOS ONE. After careful consideration, we feel that it has merit but does not fully meet PLOS ONE’s publication criteria as it currently stands. Therefore, we invite you to submit a revised version of the manuscript that addresses the points raised during the review process.

Please read the reviewer comments given below

Please submit your revised manuscript by12th August 2021. If you will need more time than this to complete your revisions, please reply to this message or contact the journal office at plosone@plos.org. Please include the following items when submitting your revised manuscript:

We look forward to receiving your revised manuscript.

Kind regards,

Muhammad Shahzad Aslam, Ph.D.,M.Phil., Pharm-D

Academic Editor

PLOS ONE

Reviewers' comments:

Reviewer's Responses to Questions

**Comments to the Author**

1. If the authors have adequately addressed your comments raised in a previous round of review and you feel that this manuscript is now acceptable for publication, you may indicate that here to bypass the “Comments to the Author” section, enter your conflict of interest statement in the “Confidential to Editor” section, and submit your "Accept" recommendation.

Reviewer #1: All comments have been addressed

Reviewer #4: (No Response)

2. Is the manuscript technically sound, and do the data support the conclusions?

Reviewer #1: Yes

Reviewer #4: Partly

3. Has the statistical analysis been performed appropriately and rigorously? 

Reviewer #1: Yes

Reviewer #4: N/A

4. Have the authors made all data underlying the findings in their manuscript fully available?

Reviewer #1: Yes

Reviewer #4: Yes

5. Is the manuscript presented in an intelligible fashion and written in standard English?

Reviewer #1: Yes

Reviewer #4: Yes

6. Review Comments to the Author

Reviewer #1: (No Response)

Reviewer #4: Thanks to the author for his availability to answer the questions asked by the reviewers. As for the manuscript, essential questions remain that still need to be clarified.

1. When re-reading the manuscript, after adjustments, it is not clear to the reader the reason for analyzing only hospital expenses and only expenses in capitals and DF. For most cancers, including breast cancer, ambulatory (outpatient) costs are much higher than hospital (inpatient) costs. Why not analyze both expenses? Why analyze only spending in capitals and DF? Wouldn't the expenses in the UF be more significant than in the capitals? Was the choice made for greater ease, convenience, or the availability in data collection, or is there any other reason?

2. The usefulness of presenting breast cancer incidence (number and raw rate) and hospitalization (number and age-standardized rate) data are not convincing. Where was this data used in the manuscript? What is the need for four tables to present unused data?

3. It is unclear what is the usefulness of using the HAQ Index. There is no mention of a theoretical or empirical association between this indicator and the prevalence of physical inactivity in the introduction. What premise or assumption motivated the author to include this analysis?

4. In the reference article by Kyu et al. (2016), the RR varies according to activity level. Compared with less active women (reporting less than 600 MET minutes/week of total physical activity), the risk of breast cancer among those in the low active (600-3999 MET minutes), moderately active (4000-7999 MET minutes), and highly active (≥8000 MET minutes) categories were, respectively, reduced by 3%, 6%, and 14%. The author used Vigitel's physically inactive women as a category of prevalence analysis. This category corresponds to how many METs/minutes/week? The impression is that the correct category of choice for Vigitel would be "insufficient physical activity" and not "physically inactive." Clarify this question, please.

5. According to the 2018 WCRF report on physical activity and cancer risk, physical activity has been associated with postmenopausal breast cancer (C50 for women 50 years or older). Why were all women included?

6. The methodology section needs to be better organized and written. The suggestion is to divide the section into four subsections: study design, hospital cost data, RR and prevalence of physical inactivity, and data analysis. The study design is a cost-of-illness study to estimate the direct costs of breast cancers attributable to lack of physical activity from the perspective of the Brazilian SUS. This approach uses aggregated disease costs, and attributable population fraction (PAF) estimates to calculate the costs attributable to a given risk factor. Hospital cost data should inform the data source and how they were collected (this information is already contained in the latest version of the manuscript). The RR and prevalence subsection should inform where the values were taken from. In the case of prevalence, the data source, the eligible population, and how the METS of the selected population were calculated need to be precise. Data analysis should inform how the PAF (equation) was calculated, the variables included in the equation, how the PAF was applied to expenses, and which software was used to calculate the final amounts.

7. Results must prioritize spending.

8. The reason for separating expenditures into 20-59 years and ≥ 60 years does not seem adequate. There are no restrictions on physical activity for those over 65s in the latest version of the WHO physical activity guide (2020). Therefore, any policy, program, or action involving physical activity will have the same recommendation regardless of premenopausal or postmenopausal women.

It is appropriate to analyze spending only in postmenopausal women since, according to WCRF, physical activity is related to postmenopausal breast cancer.

9. The discussion and conclusion sections should be rewritten in light of the above recommendations.

7. PLOS authors have the option to publish the peer review history of their article (what does this mean?). If published, this will include your full peer review and any attached files.

Reviewer #1: No

Reviewer #4: No

---

## [Author Response · Author response to Decision Letter 1]

11 Aug 2021

Reviewer #4:

Thanks to the author for his availability to answer the questions asked by the reviewers. As for the manuscript, essential questions remain that still need to be clarified.

Authors: We really appreciate your comments. In the new version of the article we have added most of your suggestions. In other suggestions, we provide explanations based on the literature to exemplify our decisions.

Reviewer #4:

1. When re-reading the manuscript, after adjustments, it is not clear to the reader the reason for analyzing only hospital expenses and only expenses in capitals and DF. For most cancers, including breast cancer, ambulatory (outpatient) costs are much higher than hospital (inpatient) costs. Why not analyze both expenses? Why analyze only spending in capitals and DF? Wouldn't the expenses in the UF be more significant than in the capitals? Was the choice made for greater ease, convenience, or the availability in data collection, or is there any other reason?

Authors: We really appreciate your comments. In the new version of the article we added outpatient costs. Thus, this article presents hospitalization costs and outpatient costs. 

The decision to analyze the capitals of Brazil and the Federal District was made because the survey of physical activity that is carried out in Brazil annually (e.g., VIGITEL) is carried out in Brazilian capitals and in the Federal District. Therefore, making estimates for all municipalities in Brazil based on physical activity information from Brazilian capitals would not be appropriate. We emphasize that there are no annual physical activity surveys covering all Brazilian states. These surveys only cover the capitals. In the new version of the paper we added this limitation: “This study developed analyses at level of Brazilian capitals, and not at level of states. The strategy used in the present study limits the identification of which states or other cities of the Brazil have high hospitalization and ambulatory costs due to physical inactivity and which would need to invest more in physical activity programs in primary care. The decision to analyze the capitals of Brazil and the Federal District was made because the survey of physical activity that is carried out in Brazil annually is carried out in Brazilian capitals and in the Federal District [17-19]”.

Reviewer #4:

2. The usefulness of presenting breast cancer incidence (number and raw rate) and hospitalization (number and age-standardized rate) data are not convincing. Where was this data used in the manuscript? What is the need for four tables to present unused data?

Authors: We really appreciate your comments. In the new version of the article we removed this information from the article. This information was presented as supplementary documents, as if readers want to know general information about breast cancer in Brazil, they can consult the supplementary files.

Reviewer #4:

3. It is unclear what is the usefulness of using the HAQ Index. There is no mention of a theoretical or empirical association between this indicator and the prevalence of physical inactivity in the introduction. What premise or assumption motivated the author to include this analysis?

Authors: We agree with the reviewer. In the new version of the article we removed the information about HAQ Index. This information really was without theoretical support.

Reviewer #4:

4. In the reference article by Kyu et al. (2016), the RR varies according to activity level. Compared with less active women (reporting less than 600 MET minutes/week of total physical activity), the risk of breast cancer among those in the low active (600-3999 MET minutes), moderately active (4000-7999 MET minutes), and highly active (≥8000 MET minutes) categories were, respectively, reduced by 3%, 6%, and 14%. The author used Vigitel's physically inactive women as a category of prevalence analysis. This category corresponds to how many METs/minutes/week? The impression is that the correct category of choice for Vigitel would be "insufficient physical activity" and not "physically inactive." Clarify this question, please.

Authors: We really appreciate your comments. In the new version of the article we added this information in the method section: “... Physically inactive were subjects who did not practice any free-time physical activity in the last three months of the interview and those who did not make intense physical efforts at work, did not commute to work or school and were not responsible for heavy cleaning of the house, as recommended in literature (e.g., they are those who have energy expenditure < 600 METS minutes/week) [22]”.

In 2017, a consensus was released (Tremblay et al., 2017) in which the concepts related to physical activity and sedentary behavior were defined. In this consensus was defined that physical inactivity is an insufficient physical activity level to meet present physical activity recommendations (i.e., < 600 METS minutes/week). For that reason, we used the term physical inactivity in our paper. 

Reference:

Tremblay MS, Aubert S, Barnes JD, Saunders TJ, Carson V, Latimer-Cheung AE, Chastin SFM, Altenburg TM, Chinapaw MJM; SBRN Terminology Consensus Project Participants. Sedentary Behavior Research Network (SBRN) - Terminology Consensus Project process and outcome. Int J Behav Nutr Phys Act. 2017 Jun 10;14(1):75. doi: 10.1186/s12966-017-0525-8.

Reviewer #4:

5. According to the 2018 WCRF report on physical activity and cancer risk, physical activity has been associated with postmenopausal breast cancer (C50 for women 50 years or older). Why were all women included?

Authors: In the new version of the paper we analyzed information from adult and older adult women in Brazil (aged ≥ 20 years) without stratifying by age. I respect the reviewer's opinion that the WCRF reports that physical activity is related to postmenopausal breast cancer. However, in the present study we used a reference for PAF that provided evidence for the female population over 20 years of age. In addition, the guidelines of physical activity make it clear that there is sufficient evidence of the relationship between physical activity and breast cancer in women adults and older adults (before and after menopause). For this reason, in the present study we made this methodological decision. The following paragraph was added to the discussion section: “There is a lot of debate in the literature about the effect of physical activity on the prevention of breast cancer before and after menopause [39-41]. All health agencies reinforce that the effect of physical activity in preventing breast cancer is more evident after menopause. However, before menopause there is a theoretical discussion that still needs consensus. The World Cancer Research Fund report describes that the strongest evidence that exists in the relationship between physical activity and breast cancer prevention is for women after menopause [39]. Physical activity guidelines describe that there is strong evidence that physical activity prevents breast cancer before and after menopause [40, 41]. The present study adopted the recommendations of the physical activity guidelines because they were the most recent on the subject. In addition, in the present study was used a reference for PAF that which investigated adult and older adults women [24]”.

Reviewer #4:

6. The methodology section needs to be better organized and written. The suggestion is to divide the section into four subsections: study design, hospital cost data, RR and prevalence of physical inactivity, and data analysis. The study design is a cost-of-illness study to estimate the direct costs of breast cancers attributable to lack of physical activity from the perspective of the Brazilian SUS. This approach uses aggregated disease costs, and attributable population fraction (PAF) estimates to calculate the costs attributable to a given risk factor. Hospital cost data should inform the data source and how they were collected (this information is already contained in the latest version of the manuscript). The RR and prevalence subsection should inform where the values were taken from. In the case of prevalence, the data source, the eligible population, and how the METS of the selected population were calculated need to be precise. Data analysis should inform how the PAF (equation) was calculated, the variables included in the equation, how the PAF was applied to expenses, and which software was used to calculate the final amounts.

Authors: The change was made. 

Reviewer #4:

7. Results must prioritize spending.

Authors: In the new version of the paper we prioritize spending information.

Reviewer #4:

8. The reason for separating expenditures into 20-59 years and ≥ 60 years does not seem adequate. There are no restrictions on physical activity for those over 65s in the latest version of the WHO physical activity guide (2020). Therefore, any policy, program, or action involving physical activity will have the same recommendation regardless of premenopausal or postmenopausal women.

It is appropriate to analyze spending only in postmenopausal women since, according to WCRF, physical activity is related to postmenopausal breast cancer.

Authors: In the new version of the paper we analyzed information from adult and older adult women in Brazil (aged ≥ 20 years) without stratifying by age. I respect the reviewer's opinion that the WCRF reports that physical activity is related to postmenopausal breast cancer. However, in the present study we used a reference for PAF that provided evidence for the female population over 20 years of age. In addition, the guidelines of physical activity make it clear that there is sufficient evidence of the relationship between physical activity and breast cancer in women adults and older adults (before and after menopause). For this reason, in the present study we made this methodological decision. The following paragraph was added to the discussion section: “There is a lot of debate in the literature about the effect of physical activity on the prevention of breast cancer before and after menopause [39-41]. All health agencies reinforce that the effect of physical activity in preventing breast cancer is more evident after menopause. However, before menopause there is a theoretical discussion that still needs consensus. The World Cancer Research Fund report describes that the strongest evidence that exists in the relationship between physical activity and breast cancer prevention is for women after menopause [39]. Physical activity guidelines describe that there is strong evidence that physical activity prevents breast cancer before and after menopause [40, 41]. The present study adopted the recommendations of the physical activity guidelines because they were the most recent on the subject. In addition, in the present study was used a reference for PAF that which investigated adult and older adults women [24]”.

9. The discussion and conclusion sections should be rewritten in light of the above recommendations.

Authors: We really appreciate your comments on the article. The change was made.

---

## [Decision Letter · Decision Letter 2]

29 Sep 2021

PONE-D-21-09895R2Hospitalization and ambulatory costs related to breast cancer due to physical inactivity in the Brazilian state capitalsPLOS ONE

Dear,

Thank you for submitting your manuscript to PLOS ONE. After careful consideration, we feel that it has merit but does not fully meet PLOS ONE’s publication criteria as it currently stands. Therefore, we invite you to submit a revised version of the manuscript that addresses the points raised during the review process. 

We look forward to receiving your revised manuscript.

Kind regards,

Muhammad Shahzad Aslam, Ph.D.,M.Phil., Pharm-D

Academic Editor

PLOS ONE

Journal Requirements:

Additional Editor Comments (if provided):

Reviewers' comments:

Reviewer's Responses to Questions

**Comments to the Author**

1. If the authors have adequately addressed your comments raised in a previous round of review and you feel that this manuscript is now acceptable for publication, you may indicate that here to bypass the “Comments to the Author” section, enter your conflict of interest statement in the “Confidential to Editor” section, and submit your "Accept" recommendation.

Reviewer #4: All comments have been addressed

Reviewer #7: (No Response)

Reviewer #8: (No Response)

2. Is the manuscript technically sound, and do the data support the conclusions?

Reviewer #4: Yes

Reviewer #7: Partly

Reviewer #8: Yes

3. Has the statistical analysis been performed appropriately and rigorously? 

Reviewer #4: Yes

Reviewer #7: No

Reviewer #8: (No Response)

4. Have the authors made all data underlying the findings in their manuscript fully available?

Reviewer #4: Yes

Reviewer #7: Yes

Reviewer #8: No

5. Is the manuscript presented in an intelligible fashion and written in standard English?

Reviewer #4: Yes

Reviewer #7: Yes

Reviewer #8: Yes

6. Review Comments to the Author

Reviewer #4: I have only a final commentary. The author said that periodic exams prevent breast cancer (lines 315-316): "it is necessary that cities invest in other initiatives for the prevention of breast cancer, such as periodic exams." It's not true. Periodic exams such as mammography, contrary to routine cervical cancer exams that lower cervical cancer cases, increase the number of breast cancers.

Reviewer #7: The topic is of great public health importance. But it only focused on description of the situation. It should be very informative to the reader and policy makers. So, higher order statistics should be included. Sound study participants categorization should be include.

Reviewer #8: (No Response)

7. PLOS authors have the option to publish the peer review history of their article (what does this mean?). If published, this will include your full peer review and any attached files.

Reviewer #4: No

Reviewer #7: No

Reviewer #8: No

---

## [Author Response · Author response to Decision Letter 2]

28 Oct 2021

Reviewer #4:

I have only a final commentary. The author said that periodic exams prevent breast cancer (lines 315-316): "it is necessary that cities invest in other initiatives for the prevention of breast cancer, such as periodic exams." It's not true. Periodic exams such as mammography, contrary to routine cervical cancer exams that lower cervical cancer cases, increase the number of breast cancers.

Authors: We really appreciate your comments. In the new version of the article we have removed that sentence.

Reviewer #7:

The topic is of great public health importance. But it only focused on description of the situation. It should be very informative to the reader and policy makers. So, higher order statistics should be included. Sound study participants categorization should be include.

Authors: We really appreciate your comments. This is a cost-of-illness study to estimate the direct costs of breast cancers attributable to lack of physical activity from the perspective of the Brazilian health services. The study's analysis units are the states of Brazil. Thus, in the article we are not working with data from individuals, but with data from the states of Brazil. In the new version of the article, we've added this information more explicitly: “The study's analysis units are the states of Brazil”.

---

## [Decision Letter · Decision Letter 3]

23 Nov 2021

Hospitalization and ambulatory costs related to breast cancer due to physical inactivity in the Brazilian state capitals

PONE-D-21-09895R3

Dear,

We’re pleased to inform you that your manuscript has been judged scientifically suitable for publication and will be formally accepted for publication once it meets all outstanding technical requirements.

Kind regards,

Muhammad Shahzad Aslam, Ph.D.,M.Phil., Pharm-D

Academic Editor

PLOS ONE

Additional Editor Comments (optional):

Reviewers' comments:

Reviewer's Responses to Questions

**Comments to the Author**

1. If the authors have adequately addressed your comments raised in a previous round of review and you feel that this manuscript is now acceptable for publication, you may indicate that here to bypass the “Comments to the Author” section, enter your conflict of interest statement in the “Confidential to Editor” section, and submit your "Accept" recommendation.

Reviewer #4: All comments have been addressed

2. Is the manuscript technically sound, and do the data support the conclusions?

Reviewer #4: Yes

3. Has the statistical analysis been performed appropriately and rigorously? 

Reviewer #4: Yes

4. Have the authors made all data underlying the findings in their manuscript fully available?

Reviewer #4: Yes

5. Is the manuscript presented in an intelligible fashion and written in standard English?

Reviewer #4: Yes

6. Review Comments to the Author

Reviewer #4: (No Response)

7. PLOS authors have the option to publish the peer review history of their article (what does this mean?). If published, this will include your full peer review and any attached files.

Reviewer #4: No

---

## [Editor Report · Acceptance letter]

7 Jan 2022

PONE-D-21-09895R3 

Hospitalization and ambulatory costs related to breast cancer due to physical inactivity in the Brazilian state capitals 

Dear Dr. Silva:

I'm pleased to inform you that your manuscript has been deemed suitable for publication in PLOS ONE. Congratulations! Your manuscript is now with our production department. 

Kind regards, 

on behalf of

Dr. Muhammad Shahzad Aslam 

Academic Editor

PLOS ONE